



# Observations of surface energy fluxes and meteorology in the seasonally snow-covered high-elevation East River Watershed during SPLASH, 2021-2023

Christopher J. Cox[1], Janet M. Intrieri[1], Brian Butterworth[1,2], Gijs de Boer[1,2,3], Michael R. Gallagher[1,2],
Jonathan Hamilton[1,2], Erik Hulm[4], Tilden Meyers[5], Sara M. Morris[6], Jackson Osborn[1], P. Ola G.
Persson[1,2], Benjamin Schmatz[4], Matthew D. Shupe[1,2], James M. Wilczak[1]

[1]NOAA Physical Sciences Laboratory (PSL), Boulder, CO 80305, USA
[2]Cooperative Institute for Research in Environmental Sciences (CIRES), University of Colorado, Boulder, CO 80309, USA
[3]Integrated Remote and In Situ Sensing (IRISS), University of Colorado, Boulder, CO 80309, USA
[4] Rocky Mountain Biological Laboratory (RMBL), Crested Butte CO 81224, USA
[5]NOAA Air Resources Laboratory (ARL), Oak Ridge, TN 37830, USA
[6]NOAA Global Monitoring Laboratory (GML), Boulder, CO 80305, USA

*Correspondence to*: Christopher J. Cox (christopher.j.cox@noaa.gov)

**Abstract.** From autumn 2021 through summer 2023, scientists from the National Oceanic and Atmospheric Administration (NOAA) and partners conducted the Study of Precipitation, the Lower Atmosphere, and Surface for Hydrometeorology (SPLASH) campaign in the East River Watershed of Colorado. One objective of SPLASH was to observe the transfer of energy between the atmosphere and the surface, which was done at several locations. Two remote sites were chosen that did not have access to power utilities. These were along the valley floor near the East River in the vicinity of the unincorporated town of Gothic, Colorado. Energy balance measurements were made at these locations using autonomous, single-level flux towers referred to as Atmospheric Surface Flux Stations (ASFS). The ASFS were deployed on 28 September 2021 at the "Kettle Ponds Annex" site and on 12 October 2021 at the "Avery Picnic" site and operated until 19 July and 21 June 2023, respectively. Measurements included basic meteorology; upward and downward longwave and shortwave radiative fluxes, and subsurface conductive flux, each at 1-minute resolution; 3-d winds from a sonic anemometer and $H_2O/CO_2$ from an open-path gas analyser, both at 20 Hz from which sensible, latent heat, and $CO_2$ fluxes were derived; and profiles of soil properties in the upper 0.5 m (both sites) and temperature profiles through the snow (at Avery Picnic), each reported between 10 min and 6 hours. For most measurements, uptime was 96% (Kettle Ponds) and 89% (Avery Picnic), and collectively 1,184 days of data were obtained between the stations. The purpose of this manuscript is to document the ASFS deployment at SPLASH, the data acquisition and post-processing of measurements, and to serve as a guide for interested users of the data sets, which are archived under the Creative Commons 4.0 Public Domain licensing at Zenodo.



## 1 Introduction

The Upper Colorado River Basin (UCRB), which spans parts of Colorado, Utah, Wyoming, Arizona, and New Mexico, is a snow dominated watershed. The water stored there in mountain snowpack supplies reservoirs used to support communities and industry throughout the southwest United States. In turn, management of these reservoirs of water depends on the fidelity of measurements and predictions of hydrological inputs and timing of discharge within the riverine system. To support advancement of the tools used for these predictive services, the National Oceanic and Atmospheric Administration (NOAA) supported a campaign to make detailed observations pertaining to snowpack, soil, and precipitation in the UCRB from 2021 to 2023, termed the Study of Precipitation, the Lower Atmosphere, and Surface for Hydrometeorology (SPLASH) (de Boer et al., 2023). SPLASH was carried out in the East River Watershed, an approximately 300 km$^2$ high-altitude area along the western slope of Colorado's Rocky Mountains near Crested Butte. The valley is additionally the focus of a U.S. Department of Energy (DoE) Watershed Function Project Science Focus Area (Hubbard et al., 2018) managed by the Lawrence Berkeley National Laboratory (LBNL) and is home to the Rocky Mountain Biological Laboratory (RMBL).

The goals of the two-year SPLASH campaign were to collect data suitable for analysing subgrid variability and physical process representation in models such as the National Water Model (NWM) and the High-Resolution Rapid Refresh (HRRR) forecast model. Of particular interest was the seasonal water cycle in complex terrain from the lower atmosphere through the upper soil. SPLASH included observations made by a network of radars, aircraft, and *in situ* stations, and was coordinated jointly with other complementary programs, including the Surface Atmosphere Integrated Laboratory (SAIL), which was operated by the DoE Atmospheric Radiation Measurement (ARM) program (Feldman et al., 2023) and the Sublimation of Snow (SOS) experiment (Gutmann et al., 2023), supported by the National Science Foundation. The general scope and coordinating activities amongst the programs and with SPLASH's host institution, RMBL, are described by the referenced studies.

The purpose of this manuscript is to provide details of the deployment, data acquisition, post-processing, data organization, and availability of one component of SPLASH, the Atmospheric Surface Flux Stations (ASFS). The ASFS contribution was supported by the NOAA Physical Sciences Laboratory (PSL) and the Cooperative Institute for Research in Environmental Sciences (CIRES). The ASFS are autonomous, single-level flux towers designed to observe energy exchanges between the atmosphere and surface at sufficiently high resolution for supporting process-based research on the coupling between these layers. The ASFS observations have been post-processed and quality-controlled, and have been archived at Zenodo under the Creative Commons 4.0 Public Domain licensing. We begin by contextualizing the observations within the surface energy budget equation in Sect. 2. In Sect. 3, we describe the ASFS system and deployment followed in Sect. 4 by a description of



the deployment sites. Sect. 5 contains details on data post-processing and uncertainties and we conclude in Sect. 6 with information about data availability and formatting.

## 2. Surface energy balance

The SPLASH observations described here focus on the terms necessary to solve the net surface heat flux equation, which represents the balance of vertical radiative, conductive, and turbulent energy transfer across the infinitely thin plane of the surface such that

$$Q = H_s + H_l + C, \tag{1}$$

where $H_s$ is the turbulent sensible heat flux, $H_l$ is the turbulent latent heat flux, $C$ is the soil conductive flux at the surface, and
$Q$ is the net surface radiation, being defined as follows:

$$Q = (SW_D - SW_U) + (LW_D - LW_U), \tag{2}$$

where $D$ and $U$ refer to downwelling and upwelling components of the shortwave ($SW$) and longwave ($LW$) radiative fluxes. $Q$ is defined as positive warming the surface while $H_s$, $H_l$, and $C$ are positive cooling the surface, as is the typical convention (these are also the conventions used in the data set).


In practice, observations of Eq. (1) frequently do not balance (e.g., Foken et al., 2006) and thus it is more practical to formulate Eq. (1) including a residual, $R$, such that

$$Q = H_s + H_l + C + R. \tag{3}$$

Grachev et al. (2020) provides a thorough analysis and review of $R$ and its implications for terrestrial applications in
balancing Eq. (3). We encourage readers to consult that resource for more details while our purpose here is to contextualize the observations we made during SPLASH. In keeping with Grachev et al.,

$$R = T + S + X, \tag{4}$$

where $T$ represents unaccounted horizontal and vertical transports (e.g., from spatial heterogeneity); $S$ is the storage in the vertical column defined by the instrument locations relative to the surface, which in practice does not meet the infinitely thin
requirement of Eq. (1); and $X$ represents a variety of sources of uncertainty in the terms comprising Eq. (3). $S$ can be further subdivided into sources. Again, following Grachev et al.,

$$S = S_a + S_g + S_p + S_c + S_x, \tag{5}$$

where $S_a$ represents flux divergence within the air column between the surface and the actual measurement heights; $S_g$ is storage of energy in the ground between the surface and the subsurface conductive flux measurement; $S_p$ is the
photosynthetic heat storage; $S_c$ is a vegetation biomass heat storage term; and $S_x$ is all other terms, such as the heat flux from precipitation (e.g., He et al., 2023) and the latent energy associated with snowmelt. For the purposes of the present study, we further subdivide $S_g$ into components of the soil ($S_{soil}$) and snow ($S_{snow}$) layers thusly,



$$S_g = S_{soil} + S_{snow}. \tag{6}$$

While the purpose of the present study is not to provide a discourse on these many details, it is practical to provide the
equations because it permits us to be more precise in defining the measurements that were made and their associated
processing, as well as to highlight those terms that have not yet been constrained or for which relevant measurements were
not made.

## 3. ASFS

The ASFS were initially designed and used for observing the energy exchange over sea ice in the central Arctic Ocean at
unstaffed locations accessed only periodically during the Multidisciplinary drifting Observatory for the Study of Arctic Climate
(MOSAiC) (Cox et al., 2023a; Shupe et al., 2022). The device (Figures 1, 2) is an aluminium pipe structure approximately 3
m long and 2 m high, set on skis similar to a rigid Nansen sled on which an electronics box is mounted. From the main structure
protrudes a horizontal boom 3 m outward from the front at a height of 2 m from the surface and a vertical mast 3 m high, both
for mounting instrumentation. The station is powered using an EFOY Pro Duo 2400 Direct Methanol Fuel Cell supplied by
120 L of methanol, which generates the 60-70 W required for typical use for up to 2 months between refuelling. The
instrumentation suite (Table 1) includes basic meteorology, sensors for detecting surface temperature and height (for snow
depth), a sonic anemometer, an open-path gas analyser, a four-component broadband radiation suite, a GPS, soil heat flux
plates, and a soil temperature and moisture probe. The data acquisition and communications are managed by a Campbell
Scientific CR1000X data logger.

The details of ASFS use and post-processing of data during MOSAiC are described in Cox et al. (2023a). Two ASFS built for
MOSAiC were repurposed for SPLASH. The stations are optimized for cold climates and perform well maintaining ice-free
instrumentation (Cox et al. 2021; Cox et al. 2023a). There are, however, several aspects of the UCRB alpine environment that
differ from Arctic sea ice in ways that were significant for the SPLASH deployment. While the systems were largely unchanged
for SPLASH, some adjustments were made:
(1) The stations were originally designed for a snow depth of < 0.5 m, typical of the Arctic. For SPLASH, the stations
needed to be raised to accommodate up to 2 m of snow. Instead of a costly redesign of the structure, we opted to
position the sleds on top of wooden picnic tables, which increased the height of the boom to 2.75 m. While simplistic,
this arrangement was found to be robust: we did not need to reseat any of the equipment and the level measured at
the mast was maintained within 1°. Due to winter accumulation of snow, the effective measurement height of the
instrumentation changed over time.
(2) The stations are capable of direct peer-to-peer radio communications (2.4 GHz) as a primary form of data transfer to
a base station. At SPLASH, without line-of-sight in the mountainous terrain, the data were stored locally on microSD
cards and retrieved manually during refuelling visits, approximately monthly. Only infrequent (3 hour) samples and





diagnostics were transferred daily to the base station for monitoring purposes (displayed on the NOAA/PSL website, https://psl.noaa.gov/splash/) using the station's back-up (Iridium dial-up data) satellite link communications.

(3)   At MOSAiC, complementary subsurface measurements were collected using buoys deployed by collaborators (Nicolaus et al. 2022, Rabe et al. 2022). To fill this gap for SPLASH, a 0.5 m soil probe was added to the ASFS instrument suite to collect soil temperature and water content information. In year two, an experimental thermistor
string for measuring temperature profiles through snow was also added to one station. Details of these sensors are provided later in the manuscript.

(4)   The EFOY is most efficient at sea level and specified to operate up to 1500 m in altitude. Additionally, while it cannot be started from a frozen state, it performs best in the cold and can overheat easily in its insulated box. The SPLASH domain has warm, snow-free summers and is near 3000 m elevation. Thus, we anticipated a decrease in efficiency in
both harvesting energy and ventilation. While we do not have objective measures, our experience was that the systems fuel burn rate was similar to that at sea level. We improved the ventilation but still experienced several instances of overheating in summer. Overall, the lifetime of the fuel cells was shorter than expected and some instruments had to be turned off to conserve power and extend the life of the system (described later).

## 4. Sites

The East River is located north of Gunnison, Colorado, along the western slopes of the Rocky Mountains. It flows southeast from Emerald Lake (3186 m) ~62 km to a confluence (2446 m) with the Taylor River near Almont, forming the Gunnison River, a tributary to the Colorado River. Gothic (2895 m), an unincorporated town that is home to RMBL, is located near the headwaters of the East River at the base of Gothic Mountain (3850 m). The slopes of the subalpine valley near Gothic include mixed stands of aspen, subalpine fir, and Engelmann spruce, while few trees are found in the valley floor where fescue grasses
and sagebrush dominate (Barry et al., 2001). The area around Gothic is transiently ranch land during the shoulder seasons as cattle are moved between the high elevation and down-valley rangelands. The ASFS were installed in this valley floor environment. Once installed, the stations operated largely autonomously. Regular maintenance visits were made every 4-6 weeks to retrieve data and refuel, and irregular visits were made as needed to conduct repairs and collect samples. Maps and a photograph of the valley annotated with the locations of all SPLASH sites can be found in Figure 1 from de Boer et al. (2023).
More detailed information about the sites specific to the present manuscript can be found in Figures 1 and 2.

### 4.1. Kettle Ponds Annex

A high density of measurements from SPLASH, SAIL, and SOS, were located at the Kettle Ponds Site ("KPS" in de Boer et al. 2023), ~2.25 km southeast from Gothic ("GTH" in de Boer et al., 2023). Kettle Ponds Site Annex (KPS-A) was situated on a parcel of land owned by RMBL approximately 400 m south (and down-valley) from KPS. ASFS station #30 ("ASFS-30"),
installed at 38°56.3686' N, 106°58.1781' W on 28 September 2021, was the only equipment stationed at KPS-A. ASFS-30



operated until 19 July 2023. KPS-A lies within a wide, gently sloping portion of the valley, locally 4-7 degrees (USGS, 2023) (Figure 1a-c). The elevation at the height of the station boom was 2856 ±0.6 m. In addition to the slope southward along the main trajectory of the valley, there is significant local component of the slope westward (Figure 1c) toward the East River, which is ~60 m lower in elevation than KPS-A. The nearest of the kettleholes that form the namesake for the site is located

~180 m upslope. Photographs of the station can be found in Figure 1d-f.

ASFS-30 was installed on two picnic tables with the boom facing approximately south to reduce shading of the radiation sensors at the end of the boom. Note that the direction the boom points is defined as "station north" with respect to the relative wind direction. Conductive flux plates were installed approximately 3.2 m offset from the centerline of the boom on either

side of the station at 5 cm depth in the soil. A soil probe was installed approximately 4 m off centerline of the boom to the east. The boom height over the snow-free surface as measured by snow height sensor was 288 cm. Fencing was installed during the summer at a radius of approximately 8 m from the end of the boom to protect the instrumentation from cattle.

Surface soil samples were collected and soil properties were measured from the immediate area around the station (Table 2).

A texture analysis was performed by the Colorado State University Soil, Water and Plant Testing Laboratory. These data were also obtained for KPS. The dry soil density of the samples at KPS-A and AYP was also calculated after the samples were dried at 205 °C for 48 hours. We obtained measurements of thermal conductivity using a handheld METER Tempos probe in the field in the vicinity of the sample locations. These measurements were converted to dry thermal conductivity using coincident, co-located measurements of soil volume water content (VWC) following the method of Peng et al. (2017). The values found

in Table 2 represent means of values obtained within the upper 10 cm of the soil on 12 June and 29 August 2023.

### 4.2. Avery Picnic

The Avery Picnic (AYP) site is approximately 1.8 km up-valley from Gothic along state road 317. The site is adjacent to the road on its west side, just a few meters above the East River, which is ~60 m further west. At AYP, ASFS station #50 ("ASFS-50") was installed at 38°58.3455' N, 106°59.8113' W on 12 October 2021 and operated until 21 June 2023. The elevation at

boom height was 2933 ±0.7 m. The terrain surrounding AYP is generally flat, and while not sloping like KPS-A, the surface is undulating in the vicinity of the meandering East River (Figure 2a,c). The valley floor is narrower than at KPS-A and the ground rises steeply to the east and west of the station at distances of ~150 – 250 m (Figure 2b). ASFS-50 and a separate snow-profiling thermistor string (described later) were the only equipment installed at AYP. Photographs of the station can be found in Figure 2d,e.


The deployment approach closely mimicked that of ASFS-30 at KPS-A with the station positioned on two picnic tables, the boom pointing south, and flux plates positioned 3.5 m off either side of boom centerline. The 0.5 m soil probe was deployed



approximately 4 m east of the rear of the station instead of near the boom because of difficultly encountered with a layer impenetrable to the auger near 0.45 m depth. The location that was chosen also had this layer slightly deeper and the probe
was offset from being flush with the mean surface by 3 cm but because of local variability in the surface the uppermost sensor was covered by soil and not exposed to air. The boom height over the snow-free surface as measured by the snow height sensor was 302 cm. Fencing was also installed around ASFS-50 at AYP in summer. The same procedure for soil sampling conducted at KPS-A was also carried out at AYP (Table 2). Additional spatial sampling of near-surface VWC using a handheld FieldScout Time Domain Reflectometry (TDR) Soil Moisture Meter were made periodically in vicinity of both KPS-A and AYP and may
be obtained from Zenodo (Intrieri et al., 2023).

## 5. Data Processing

In this section, we provide details relevant for user interpretation of the data set pertaining to the observing strategy, post-processing, and uncertainties of the measurements. Data gaps occurred both from downtime for entire stations and individual
sensors. Instrument uptimes during the experiment are displayed in Figure 3.

### 5.1. Meteorology

Air temperature, pressure, and humidity were measured at both sites using a Vaisala PTU307 probe. The sensor was affixed to the boom and the precise instrument heights (relative to snow-free ground) are found in Table 1. A detailed description of this sensor and intercomparisons can be found in Cox et al. (2023a). Post-processing of the data collected during SPLASH is
identical to that described in the earlier study. Figure 4a shows time series of air temperature as an example of the available meteorological data.

Downtime was only experienced for these systems when an entire station was down (Figure 3), which was due to overheating (summer) or (likely) ice-obstructed exhaust lines (winter). This occurred from 19 May through 2 June 2022 at KPS-A and 20
January through 21 February, 7-14 March, and 26 May through 12 June 2023 at AYP. The total uptime for the systems relative to the deployment beginning and end dates was ~96% and ~89% for KPS-A and AYP, respectively.

### 5.2. Snow Depth

Snow depth was calculated using Campbell Scientific SR-50A acoustic rangers. The sensors were mounted on the boom and averaged twelve samples of distances to the target (surface) each minute. These distances were corrected for temperature
dependencies on the speed of sound and then converted to snow depth by subtracting the measured value from the snow-free reference value reported in Sect. 3. The data was quality controlled using an internal diagnostic reported by the sensor for signal strength, and also manually by using a despiking routine. Snow depths are shown in Figures 4b,c (black lines). There is





a gap in the AYP record prior to 1 February 2022 because of a failure in the originally installed sensor (Figure 3b, Figure 4c).
Data collected during the snow-free period (not displayed in Figure 4) is preserved in the data set with a caution flag applied
because of noise associated with a poorly defined reflecting surface caused by vegetation.

**5.3. Radiation**

Broadband shortwave (285-3000 nm; $SW_D$, $SW_U$) and longwave (4.5-40 $\mu m$; $LW_D$, $LW_U$) radiation were measured using
Hukseflux SR30-D1 pyranometers and IR20-T pyrgeometers, respectively. The sensors were fixed to a common mounting
plate that could be manually adjusted to achieve level, which was measured continuously by inclinometers embedded in the
SR30s. The mean level of the upward-facing pyranometers for the duration of the experiment was 0.5±0.28° (KPS-A) and
0.11±0.28° (AYP). The sensors were heated and ventilated. They have been shown previously to perform well in self-
maintenance for ice mitigation (Cox et al., 2021; Cox et al., 2023a) and icing was not observed in the SPLASH data except for
occasional cases of snow accumulation on the upward-facing dome during active precipitation. The SR30s directional response
is < 1% (Cox et al., 2023a) and the sensor's infrared loss characteristics are < 3 W m$^{-2}$ (Wang et al., 2018). Calibrations were
factory (Table 1). The SR30s are polled digital RS485 protocol and the IR20s are analog; a 100 Ohm 0.01% reference resistor
was used to derive case temperature from the IR20s for calibration. The quality control procedure included both manual review
and automated detection of spurious data using the "QCRad" methodology (Long and Shi, 2008). The time series of net
radiation, $Q$, is shown in Figure 4d. The uptime for the radiative fluxes is similar to that of the meteorology (Figure 3).

To supplement the broadband longwave data, Apogee SI-400 infrared thermometers were mounted to the boom to help
characterize the surface temperature. The sensor at AYP performed as expected but unfortunately the one at KPS-A shows
values more characteristic of the sensor temperature than the target temperature, a feature most obvious when observing warm
temperatures over the melting snow surface in spring when the target is ~ 0 °C. These data from KPS-A are preserved in the
data set with a cautionary flag and we consider them suspect.
The surface skin temperature was derived from the pyrgeometers, accounting for emissivity and reflections, using the following
equation,

$$T_s = \left\{ \frac{LW_U - (1-\varepsilon)LW_D}{\varepsilon\sigma} \right\}^{0.25}, \tag{7}$$

where $\varepsilon$ is emissivity and $\sigma$ is the Stefan-Boltzmann constant. The emissivity was set to 0.985 when the surface was covered
by snow (Warren, 1982) and to 0.975 for grassland (Hu et al., 2019) during the snow-free period. No further attempt to
constrain the emissivity was conducted for the present data set and doing so is considered ongoing work by the SPLASH
team and collaborators.



Occasionally in spring very large gradients in near-surface temperature (> 10 °C between $T_s$ and 2 m) were observed in
association with above-freezing air being advected over the snow surface. Under these conditions, the air in the path between
the $LW_U$ sensor (mounted to the boom) and the surface (target) was sufficiently warmer than the target to bias the measurement
high due to emission at wavelengths > 14 $\mu$m where the IR20-T is sensitive and the water vapour and $CO_2$ in the atmosphere
are highly absorptive. These factors are contributions to $S_a$ from Eq. (5), which we accounted for in $LW_U$ so that the $T_s$ could
be more accurately derived. To estimate this influence, we ran a series of broadband longwave flux calculations using the
Rapid Radiative Transfer Model (RRTM) (Mlawer et al., 1997) with input from representative radiosonde profiles launched
by SAIL (Keeler et al., 2021) and specifying a range of near-surface temperature gradients between 0 and 14 °C. From the
results, we calculated the flux divergence between the surface and a nominal instrument height of 2 m over snow as a function
of the gradient, which is approximately linear in the ranges tested and of similar scale to the calibration uncertainty. A simple
linear fit is used to apply a first-order correction to the $LW_U$ based on the initially observed gradient,

$$S_{a,LW} = 0.29193 * \Delta T - 0.021642, \tag{8}$$

where $\Delta T$ is the air temperature minus an initial calculation of the surface skin temperature. Analogous contributions to $S_a$ in
$SW_D$ and $SW_U$ are considered negligible and we did not adjust $LW_D$ in the data set.

Figure 5 presents a cross-validation between the radiative fluxes (1-hour means) observed by ASFS-30 at KPS-A and those
made ~400 m up-valley at KPS by the NOAA Global Monitoring Laboratory (GML) (Soldo et al., 2023), also part of SPLASH.
The comparison is independent: the equipment, calibration procedures, maintenance, mounting, and post-processing all differ
between the GML and ASFS stations. We expect $SW_D$ to be comparable when averaged over 1-hour across the distance
between the stations with the potential exception of the timing of mountain shadows near sunrise and sunset. The results in
Figure 5a demonstrate excellent reproducibility between the systems, with a combined bias of < 1.9%, which is better than the
(combined) target uncertainty of 2.8% (McArthur, 2005). We also expect $LW_D$ to be comparable. Indeed, 69% of the
observations fall within the expected (combined) 2.8 W m$^{-2}$ 1σ difference (McArthur, 2005; Cox et al., 2021), which compares
well to a 68% nominal expectation of a normal distribution. However, approximately 9% of the observations comprise a long
tail of warm values at KPS-A relative to KPS. We do not have a definitive explanation for these differences. The two stations
were at different heights above the surface, but the differences are uncorrelated with the near-surface temperature gradient.
The differences tend to occur when the atmospheric boundary layer is well-mixed beneath thick cloud cover. We can rule out
icing of the KPS sensor (which would result in a cold bias) because the occurrences are equally likely when the temperature
is above 0 °C as below. One speculative possibility is that during precipitation events, the case heating of the IR20-T at KPS-
A increases the temperature of droplets temporarily occupying the dome surface, thereby enhancing the measured signal.
Differences in $LW_U$ (Figure 5c) and $SW_U$ (Figure 5d) are not easily separable from actual heterogeneity in the surface, but are
shown for completeness. The comparison of $SW_U$ shows a systematic difference of ~+8% at KPS-A relative to KPS, which
appears only in winter (thus are not related to differences in vegetation). We speculate that the difference is explainable by
reflection of sunlight from snow on the nearby mountain slopes. The GML station featured a housing that shades the ~5° of



the hemisphere closest to the horizon. The purpose is nominally to prevent the direct beam from appearing in $SW_U$ near sunrise/sunset due to minor errors in level, but in the East River Valley it would have reduced the sunlight reflected from

neighbouring slopes as well. On the ASFS, the full hemisphere was observed. Note that neither approach produces a spurious result, but rather a different perspective. The GML system likely produces a better estimate of the surface albedo ($SW_U / SW_D$), a property of the surface, whereas the ASFS station likely provides a more complete estimate of the local net $SW$ ($SW_D - SW_U$), a property of the local radiation field. Regardless, the differences are overall small and could also be explained in part by spatial heterogeneity in snow morphology and surface slope.

**5.4. Soil Properties and Soil Conductive Flux**

The soil probe was a 0.5 m length Campbell Scientific SoilVUE10 TDR sensor that measures VWC, temperature, electrical conductivity, and permittivity at 5, 10, 20, 30, 40, and 50 cm depth. In addition to periods when the stations were down, the probe at KPS-A suffered a severed cable in June 2022 that could not be repaired until September (Figure 3a). As an example of data from the probes, soil temperatures are plotted in Figures 4b,c (colours). In addition to the probe, conductive flux plates

(Hukseflux HFP01) were installed in the soil 5 cm below the surface. Two plates were installed at each site (Sect. 4) but unfortunately one of the flux plates failed at KPS-A. There were two problems with this sensor; first, it was inadvertently initially a single-ended measurement (floating ground) and second, in June 2022 the sensor wire was severed and repairs were not possible. We have retained the observation prior to June 2022, which is noisy and exhibited an offset (the offset was found using periods of isothermal soil temperatures and subtracted off), but the data are given a cautionary flag and considered

suspect.

Soil conductive flux was measured in two ways. The first method uses the flux plates, which made direct measurements *in situ* and thus represent the flux at 5 cm depth. Consequently, relative to the flux at the surface interface, the flux plates have not been adjusted for storage and are best defined as $C + S_g$ (Eqs. (3) and (6)) where the relative values of $S_{soil}$ and $S_{snow}$ may be

dramatically different between winter and summer and may vary substantially throughout the day as well. We were not able to calculate $S_{soil}$ due to a lack of co-located temperature data, which we believe may have been spatially variable due to the fact that a comparison of the flux plates at AYP suggest local spatial heterogeneity of at least up to a factor of two. Calculations of $S_{snow}$ are considered ongoing work (see Sect. 5.5).

We corrected for deflection error in each of the flux plates (Sauer et al., 2007; Morgensen, 1970), which arises from differences between the conductivity of the plate (0.76 W m$^{-2}$ K$^{-1}$) and the surrounding matrix. The first step is to calculate the thermal conductivity of the soil. We began with the measured dry thermal conductivity of the soil (Sect. 4.1). Then, a time-variant conductivity was calculated based on the observations of VWC following Peng et al. (2017). For times when the soil probe



VWC was unavailable, a mean value was used and the dependent variables in the data set are given a caution flag. The soil
properties necessary to make the calculation are found in Table 2.

The second estimate of conductive flux comes from the soil probe. We calculated a layer-averaged $C$ (i.e., $C - S_{soil}$) with
reference to the upper 10 cm (KPS-A) and 7 cm (AYP) and also $S_{soil}$ for the same layer. The equations for these calculations
are:

$$C - S_{soil} = k_{eff} \frac{T_s - T_{soil}}{z}, \tag{9}$$

where $T_{soil}$ refers to the temperature at depth $z$ and $k_{eff}$ is effective soil thermal conductivity, and

$$S_{soil} = C_v z_{soil} \frac{T_s^{n+1} - T_s^{n-1} + T_{soil}^{n+1} - T_{soil}^{n-1}}{2(t_{n+1} - t_{n-1})}, \tag{10}$$

where $t$ is time at index $n$ and $C_v$ is the soil heat capacity, which was found using Eq. (10) from Abu-Hamdeh (2003) using
inputs of the soil probe's VWC and values found in Table 2.

## 5.5. Snow temperature profiles

During the second year of SPLASH (Figure 3b), a stand-alone thermistor string was added to the suite of instruments at AYP.
The sensor is manufactured by SAMS Enterprise as the Snow Ice Mass Balance Apparatus (SIMBA) (Jackson et al., 2013). It
was originally intended for use as a buoy in polar sea ice (e.g., Lei et al. 2023), but was reconfigured by the manufacturer for
terrestrial applications for the Scottish Avalanche Information Service. The terrestrial version, used for SPLASH, is 1.9 m in
length with thermistors at 2 cm spacing affixed to a ridged plastic support structure and suspended from a tripod. In addition
to ambient temperature, it also features a low-power heating cycle applied to the thermistors for which cooling rates are
reported. These rates are correlated with the thermal environment, adding information for distinguishing between air and snow
(Jackson et al., 2013). We installed the SIMBA prior to snowpack onset on 1 November 2022 approximately 3 m east of ASFS-
50 and the snow was permitted to accumulate and ablate around the sensor throughout the winter and spring (Figure 4e).

335
There were two unanticipated issues with this system. Initially, we planned to obtain profiles every 10 min, but a loss of the
configuration file on 23 December 2022 was compounded by a firmware issue on 1 January 2023 that prevented us from
accessing the data until after the snow melted in spring, resulting in profiles being collected instead every 6 hours after the
configuration changed. This same firmware problem also resulted in a loss of clock synchronization (with GPS) and
340    consequently from January through June the clock drifted. We estimate that the drift was a maximum of 80 sec, but was likely
much less. The second problem is that the snow depth in spring 2023 was much deeper than anticipated. Consequently, from
22 March to 4 April, the snow surface was over the top of the sensor. It is also notable that there was a gradient in snow depth
with deeper snow at the ASFS and shallower snow east of the ASFS where the SIMBA was located. The difference near peak
snow depth was ~ 25 cm.






The archived data set includes the temperatures and associated thermistor cooling rates. The SIMBA data may also be useful in deriving profiles of conduction, density, and $k_{eff}$ through the snow (Sledd et al., 2024), allowing for $C$ and $S_{snow}$ to be obtainable at AYP during snow-covered period in 2022-2023. This work is experimental and ongoing and, if successful, higher-order metrics will be made available as part of a future update to the data set.

**5.6. Turbulent fluxes**

Turbulent sensible ($H_s$) and latent ($H_l$) heat fluxes were derived using eddy covariance methodology. The fundamental measurements were made using a Metek uSonic-3 Cage Multipath sonic anemometer and a co-located Licor 7500-DS gas analyser. The raw data was acquired at 20 Hz after which it was aggregated to 10 Hz by averaging, rotated into earth coordinates, and then quality-controlled (despiking, physically-possible limits, sensor diagnostics). Horizontal wind speed and

direction (1, 10 min means) were calculated from these processed components. For the fluxes, integration windows used in the provided data set are reported at 10 min intervals, though in practice the calculation is performed for the nearest power-of-2 number of samples and so the actual integration period is ~13.65 min, centred; missing data within the window was filled with the median value of the window. To perform the flux calculation, u, v, and w were rotated using the Planar Fit method (Wilczak et al., 2001), which puts u and v into the streamline of the horizontal wind direction (U, V) while rotating vertical (w -> W)

velocity to the long-term mean, where the mean in this case was determined using the observations from October 2021 through June 2022. The approach is thought to provide a more robust estimate of fluxes in sloping terrain, in particular for momentum fluxes. The coefficients (Figure 6) for the method at both sites were an acceptable fit, but (as expected) it is only over the sloping meadow at KPS-A where the streamlines are substantially different from the level anemometer. Nevertheless, for consistency, the Planar Fit was used for both sites.


Power- (Welch) and cross-spectral densities were calculated using the Python "scipy" module; the FFTs were Hamming filtered and linearly detrended, and the resulting spectra were smoothed before integration (Grachev et al., 2018; Cox et al., 2023a). We also applied a correction for undesirable high-frequency (> 1 Hz) correlated noise between W' and T' that is unique to the sonics used for these ASFS and that we devised during an earlier study (Cox et al., 2023a). Shadow corrections

for the sensor head geometry are performed internally by the sensor following Wyngaard and Zhang (1985). Buoyancy fluxes were then calculated from the cospectra and corrected for moisture fluctuations to yield $H_s$. While it is possible to make the correction directly using the humidity measurements from the co-located Licor, the Licor suffered more downtime than the sonic anemometer and so instead we use the method of Schotanus et al. (1983), which is based on the Bowen ratio, and is deemed necessary for environments where the Bowen ratio < 1 (~36% of the time at SPLASH). The main downtime in the

Licor during SPLASH occurred at KPS-A (Figure 3a) where the sensor was turned off after May 2022 to conserve power due to an aging EFOY generator (see Sect. 3). Secondarily, Licor data were lost when the sensor windows were covered by water, ice, or dust.



$H_l$ and $CO_2$ mass fluxes were calculated using the gas measurements from the Licor. The mounting of the sonic anemometer

and Licor on the mast follows the recommendations of Kristensen et al. (1997), both in its horizontal (being in-line with the

dominant wind direction) and vertical offsets, to minimize losses in covariance signal at high frequencies associated with the

sensor separation. We thus do not perform a sensor separation correction in post-processing and assume the mounting geometry

to minimize the error to < 2%, as found by Kristensen et al. (relative to a strictly horizontal separation and a random wind

direction). Licor data were checked for quality using the recommended diagnostics and data was also only retained when the

optics were determined to be clean and unobstructed, as evidenced by the $CO_2$ reference signal strength exceeding 94%. The

Webb density correction (Webb et al., 1980) was applied to both gas fluxes. $H_l$ was calculated using the latent heat of

sublimation over dry snow conditions and the latent heat of vaporization during snow-free conditions and when the surface

snow temperature > -0.2 °C, indicating the likelihood of wet or melting snow (when the liquid is assumed to evaporate

preferentially to the sublimation of ice in the slurry). In addition to reporting the $CO_2$ mass flux (mg m$^{-2}$ s$^{-1}$), we also report an

inferred photosynthetic storage value ($S_p$) in W m$^{-2}$ following the formulation used by Grachev et al. (2020) over grassland in

the Columbia River Basin. Because the quality of these $CO_2$ fluxes is not easily verified (e.g., Kohsiek, 2000) nor were $CO_2$

fluxes the primary focus of our measurement strategy, we have conservatively marked all $CO_2$ data with a caution flag in the

files. In addition to the heat fluxes, other conventional variables related to stress (e.g., friction velocity, u*, drag coefficient,

$C_d$) and the Monin-Obukhov stability parameter ($z/L$) are provided. We also provide diagnostics in the files (e.g., angle of

attack) as well as the spectra and co-spectra from which the fluxes are calculated.

To supplement eddy covariance data during gaps, we also calculated turbulent fluxes using the bulk aerodynamic method of

Andreas et al. (2010). The Andreas et al. method was designed for snow-covered sea ice and has a similar form to the

"COARE" algorithm widely used over global oceans (Fairall et al., 2003). The algorithm operates iteratively to constrain a

solution satisfying co-dependence between Obukhov Length, $L$, and u*. We found that the method adapted well to snow-

covered periods at SPLASH. The primary assumptions are that air in pore spaces of surface snow is saturated (for calculating

$H_l$), and that surface roughness length ($z_0$) can be reasonably approximated. We calculated a median wintertime $z_0$ for each

station using sonic anemometer data (Andreas et al., 2010; Grachev et al., 2007; Paulson, 1970) and found it to be close to that

reported by Andreas et al. for wintertime snow in the Beaufort Sea (2.3 x 10$^{-4}$ m) and so we used this published value for the

snow-covered period.

We further adapted this method for snow-free periods. To do so, we first specified $z_0$ calculated as previously described for

summer months (2.4 x 10$^{-3}$ at AYP and 3.6 x 10$^{-3}$ at KPS-A). This was sufficient to produce useful estimates of $H_s$ (Figure

7a,b). Through much of summer, the soil surface layer is dry in the East River Valley and the assumption of saturation at the

surface valid for snow fails for calculation of $H_l$. To calculate bulk $H_l$ for snow-free conditions, we began by assuming that the

surface layer relative humidity (RH) gradient tends toward 0% and thus that $H_l$ is dominated by diabatic processes. Kim et al.



(2021) demonstrates that this is an appropriate approach based on their reformulation of the Penman-Monteith equation in terms of RH. This diabatic assumption applies best when the surface and near-surface air are in thermodynamic equilibrium and the adiabatic term is negligible. This "surface flux equilibrium" has been shown to be a sufficient approximation to estimate

evapotranspiration for daily-to-monthly means in inland continental locations using only single-level atmospheric measurements (McColl and Rigden, 2019). It's utility at sub-daily scales when non-stationarity can introduce larger adiabatic influences on fluxes remains an open-question (Kim et al., 2021), but nevertheless, we find that it reproduces eddy covariance measurements of $H_l$ sufficiently well during SPLASH (Figure 7c,d) for inclusion in the data set.

Our implementation of concepts described by Kim et al. and McColl and Rigden makes use of the same Andreas et al. bulk methodology adapted to include the conventional "alpha" ($\alpha$) approach for estimating surface specific humidity (Lee and Pielke, 1992),

$$q_{sfc} = \alpha q_{sat}(T_s). \tag{11}$$

We calculate $\alpha$ following Kondo et al. (1990),

$$\alpha = \frac{q}{q_{sat}(T_s)} + \beta \left(1 - \frac{q}{q_{sat}(T_s)}\right), \tag{12}$$

where $q/q_{sat}(T_s) \sim \text{RH}_{2m}$ in accordance with Kim et al. and McColl and Rigden. $\beta$ is a wetness parameter that is calculated using soil VWC (Eq. (7) in Lee and Pielke, 1991) and helps to account for periods when the equilibration assumption is not met due to soil moistening. Except for the period immediately after snowmelt and peak saturation during rain events, the soil is dry and $\beta << \frac{q}{q_{sat}(T_s)}$. Therefore, for periods when soil moisture data is unavailable, we set $\beta = 0$ (consistent with McColl and Rigden);

inclusion of $\beta$ increases correlation (r) between eddy covariance $H_l$ and analogous bulk estimates over snow-free surfaces by < 0.05.

**6. Conclusions, data format, and availability**

To summarize, the data set described here was collected by two single-level, autonomous Atmospheric Surface Flux Stations

(ASFS) during the Study of Precipitation, the Lower Atmosphere, and Surface for Hydrometeorology (SPLASH) campaign (de Boer et al., 2023) in the East River Watershed of Colorado. Collectively between the stations, 1,184 days of data were obtained between 2021 and 2023 that include meteorology, surface energy budget variables, and (sub)surface properties.

The data are made available in daily netCDF4 formatted files and may be downloaded from Zenodo under the Creative

Commons 4.0 Public Domain licensing. Several types of files are provided. Level 1 files (Cox et al., 2023b,c) are raw data with only technical corrections (e.g., sign conventions) and metadata applied. These level 1 files include "slow" and "fast" versions, which refer respectively to data with a native resolution of 1 min (most data) and 20 Hz (sonic anemometer and gas





analyser). These data are largely for archival purposes as it is recommended that most users use level 2 data. A separate level
1 data set was also created and made available for the SIMBA data (Cox et al., 2023d).


A processed level 2 data set (Cox et al., 2023e,f), which includes quality control, corrections, and derived variables as outlined
in this manuscript is the data set recommended for most scientific purposes. There are three types of level 2 files; "sledmet*",
"sledseb*", "sledwind10Hz*". Sledmet contains the 1 min means of all relevant variables and sledseb contains 10 min means
of the same variables as well as all the bulk and eddy covariance calculations. Sledwind10Hz contains the quality controlled,

aggregated, and rotated earth coordinate sonic anemometer and gas analyser data to facilitate users who may wish to recalculate
turbulent fluxes using other approaches. Flags containing quality control information are found in the files, denoted *_qc with
the coding for the flags encapsulated in the file's global attributes. Table 3 summarizes the file names and DOIs. These data
are found within the SPLASH Zenodo "Community" amongst complementary data from other contributing SPLASH projects.
Complementary data from SAIL can be found at the ARM archive, https://arm.gov and complementary data from SOS can be

obtained through the University of Washington, https://depts.washington.edu/mtnhydr/Pages/NewDataPages/SOSdata.html.

**Author Contributions.** CC, JI, BB, GB, MG, JH, EH, SM, JO, BS, and MS carried out the field campaign. CC, JI, BB, MG,
TM, and PP prepared the data sets. CC prepared the manuscript with input from all authors.

**Competing interests.** The authors declare no competing interests.

**Acknowledgements.** We acknowledge contributions from the Rocky Mountain Biological Laboratory, who provided logistic,
facility, and coordination support, the U.S. Forest Service who manages the land at Avery Picnic, private land owners in the
East River Valley who coordinated with field operations, the SOS, SAIL, SFA, and NOAA-GML teams, as well as all our

SPLASH collaborators. We appreciate useful conversations with Joseph Sedlar and Laura Riihimaki from NOAA-GML.

**Financial support.** Support for this work was from the NOAA Physical Sciences Laboratory and NOAA Cooperative
Agreements NA17OAR4320101 and NA22OAR4320151. The original ASFS systems were built using support from the U.S.
National Science Foundation OPP-1724551.





**Table 1. Instrumentation, sampling, and uncertainties**

| Instrument | Obs. Variable | Sampling Rate | Reported Frequency | Height [m] (snow-free) | Cal. Date | Est. Uncert. | Valid Range | Uncert. Notes |
|---|---|---|---|---|---|---|---|---|
| Vaisala PTU 307 | Air temperature | 0.2 Hz | 1 min, 10 min | 2.89 | Feb 2019 | 0.35 - 0.2 °C | one-sided (±) | reported by manufacturer, -20 - +20 °C |
| Vaisala PTU 307 | Air Relative Humidity | 0.2 Hz | 1 min, 10 min | 2.6 | Feb 2019 | 1.2-1.8% | one-sided (±) | reported by manufacturer 25-100% RH, -20 - +40 °C |
| Vaisala PTU 307 | Air Pressure | 0.2 Hz | 1 min, 10 min | 2.79 | Feb 2019 | 0.3 Pa | one-sided (±) | reported by manufacturer, accounts for stability (drift) |
| Metek uSonic-3 | Wind Speed | 20 Hz | 1 min, 10 min | 4.62 | Verified July 2021 | 0.15 m s$^{-1}$ | one-sided (±) | component error reported by manufacturer |
| Metek uSonic-3 | Wind Direction | 20 Hz | 1 min, 10 min | 4.62 | Verified July 2021 | ~2° | max error | dominated by mount alignment error |
| Hukseflux SR30-D1 | $SW_{\downarrow}$ | 0.2 Hz | 1 min, 10 min | ~3 | Feb 2019 | 2% | one-sided (±) | target (McArthur, 2005). Validated Sect. 5.3, Cox et al. 2021) |
| Hukseflux SR30-D1 | $SW_{\uparrow}$ | 0.2 Hz | 1 min, 10 min | ~3 | Feb 2019 | 2% | one-sided (±) | target (McArthur, 2005). Validated Sect. 5.3, Cox et al. (2021) |
| Hukseflux IR20 | $LW_{\downarrow}$ | 0.2 Hz | 1 min, 10 min | ~3 | Mar 2019 | 2.6 W m$^{-2}$ | ±1σ | empirical, Cox et al. (2021); mean 4 W m$^{-2}$ - 1 W m$^{-2}$ (clear-cloudy) |
| Hukseflux IR20 | $LW_{\uparrow}$ | 0.2 Hz | 1 min, 10 min | ~3 | Mar 2019 | 1 W m$^{-2}$ | ±1σ | cloudy value $LW_{\downarrow}$ as estimate |
| Metek uSonic-3 | $H_s$ | 10 Hz | 10 min | 4.62 | Verified July 2021 | 4.8 W m$^{-2}$ | RMSE | sensor repeatability only (Cox et al., 2023a) |
| Licor 7500DS/Metek Hukseflux HFP01 | $H_l$ | 10 Hz | 10 min | 4.27 | Verified July 2021 | 10% | one-sided (±) | Grachev et al. (2018) |
| Hukseflux HFP01 | G (5 cm soil depth) | 0.2 Hz | 1 min, 10 min | -0.05 | Apr 2018, Jan 2019 | 6.10% | 95% CI | ignores deflection error, which is corrected. Includes stability (drift) |
| SoilVUE10 | VWC | 30 min (yr 1), 15 min (yr 2) | 1 min, 10 min | -0.5 to -0.05 | 2021 | 1.50% | one-sided (±) | reported by manufacturer |
| SoilVUE10 | Soil Temperature | 30 min (yr 1), 15 min (yr 2) | 1 min, 10 min | -0.5 to -0.05 | 2021 | 0.15 °C | one-sided (±) | reported by manufacturer |
| SR50 | Surface height | 0.2 Hz | 1 min, 10 min | ~3 | 2018 | 3 cm | ±2σ | reported by manufacturer + est. error from air temp |
| Apogee IRT | Surface brightness temp | 0.2 Hz | 1 min, 10 min | ~3 | May 2018 | 6 °C | 95% CI | reported by manufacturer + est. stability error |




**Table 2. Soil properties obtained from Avery Picnic (AYP), Kettle Ponds Annex (KPS-A) and Kettle Ponds (KPS).**

|  | KPS-A | AYP | KPS |
|---|---|---|---|
| Sand Fraction [%] | 35 | 40 | 43 |
| Silt Fraction [%] | 42 | 35 | 35 |
| Clay Fraction [%] | 22 | 42 | 22 |
| Soil Texture | Loam | Loam | Loam |
| Dry soil density ($\rho_b$) [g cm$^{-3}$] | 1.14 | 1.03 | - |
| Dry thermal conductivity [W m$^{-2}$ K$^{-1}$] | 0.21 | 0.62 | - |










**Table 3. List of file names and DOIs for levels 1 and 2 data sets.**


| Name | Level | Content | Format | Zenodo DOI | Reference |
|---|---|---|---|---|---|
| ASFS 30 measurements, KPS-A | 1 (raw) | 1 min mean | slow.sled.level1.asfs30-pond.YYYYMMDD.hhmmss.nc | https://doi.org/10.5281/zenodo.10307825 | Cox et al. (2023b) |
| ASFS 30 measurements, KPS-A | 1 (raw) | 20 Hz smp | fast.sled.level1.asfs30-pond.YYYYMMDD.hhmmss.nc | https://doi.org/10.5281/zenodo.10307825 | Cox et al. (2023b) |
| ASFS 30 measurements, KPS-A | 1 (raw) | 15 min smp | soil.sled.level1.asfs30-pond.YYYYMMDD.hhmmss.nc | https://doi.org/10.5281/zenodo.10307825 | Cox et al. (2023b) |
| ASFS 50 measurements, AYP | 1 (raw) | 1 min mean | fast.sled.level1.asfs50-picnic.YYYYMMDD.hhmmss.nc | https://doi.org/10.5281/zenodo.10310520 | Cox et al. (2023c) |
| ASFS 50 measurements, AYP | 1 (raw) | 20 Hz smp | slow.sled.level1.asfs50-picnic.YYYYMMDD.hhmmss.nc | https://doi.org/10.5281/zenodo.10310520 | Cox et al. (2023c) |
| ASFS 50 measurements, AYP | 1 (raw) | 15 min smp | soil.sled.level1.asfs50-picnic.YYYYMMDD.hhmmss.nc | https://doi.org/10.5281/zenodo.10310520 | Cox et al. (2023c) |
| SIMBA measurements, AYP | 1 (raw) | 10 min - 6 hour smp | tempprofile.sled.level1.YYYYMMDD.hhmmss.nc | https://doi.org/10.5281/zenodo.10327409 | Cox et al. (2023d) |
| ASFS 30 measurements, KPS-A | 2 (proc.) | 1 min mean | sled.met.asfs30.level2.0.1min.YYYYMMDD.hhmmss.nc | https://doi.org/10.5281/zenodo.10313894 | Cox et al. (2023e) |
| ASFS 30 measurements, KPS-A | 2 (proc.) | 10 Hz smp | sled.wind10hz.asfs30.level2.0.YYYYMMDD.hhmmss.nc | https://doi.org/10.5281/zenodo.10313894 | Cox et al. (2023e) |
| ASFS 30 measurements, KPS-A | 2 (proc.) | 10 min mean & turb. | sled.seb.asfs30.level2.0.10min.YYYYMMDD.hhmmss.nc | https://doi.org/10.5281/zenodo.10313894 | Cox et al. (2023e) |
| ASFS 50 measurements, AYP | 2 (proc.) | 1 min mean | sled.met.asfs50.level2.0.1min.YYYYMMDD.hhmmss.nc | https://doi.org/10.5281/zenodo.10313363 | Cox et al. (2023f) |
| ASFS 50 measurements, AYP | 2 (proc.) | 10 Hz smp | sled.wind10hz.asfs50.level2.0.YYYYMMDD.hhmmss.nc | https://doi.org/10.5281/zenodo.10313363 | Cox et al. (2023f) |
| ASFS 50 measurements, AYP | 2 (proc.) | 10 min mean & turb. | sled.seb.asfs50.level2.0.10min.YYYYMMDD.hhmmss.nc | https://doi.org/10.5281/zenodo.10313363 | Cox et al. (2023f) |

Earth System Open Access Science Discussions Data

**Figure 1: The Kettle Ponds Site – Annex (KPS-A). (a) Google Earth™ (Landsat/Copernicus) looking northward (upvalley). (b) and (c) are slope and aspect maps of the KPS-A vicinity (USGS, 2023). Black dots denote the location of ASFS-30. (d)-(f) are photos of ASFS-30 in February 2023, June 2023, and September 2021, respectively.**



**Figure 2: The Avery Picnic site (AYP). (a)-(c) as in Figure 1. Black dots denote the location of ASFS-50. (d), (e) are photos of ASFS-50 in January 2022 and October 2021, respectively.**


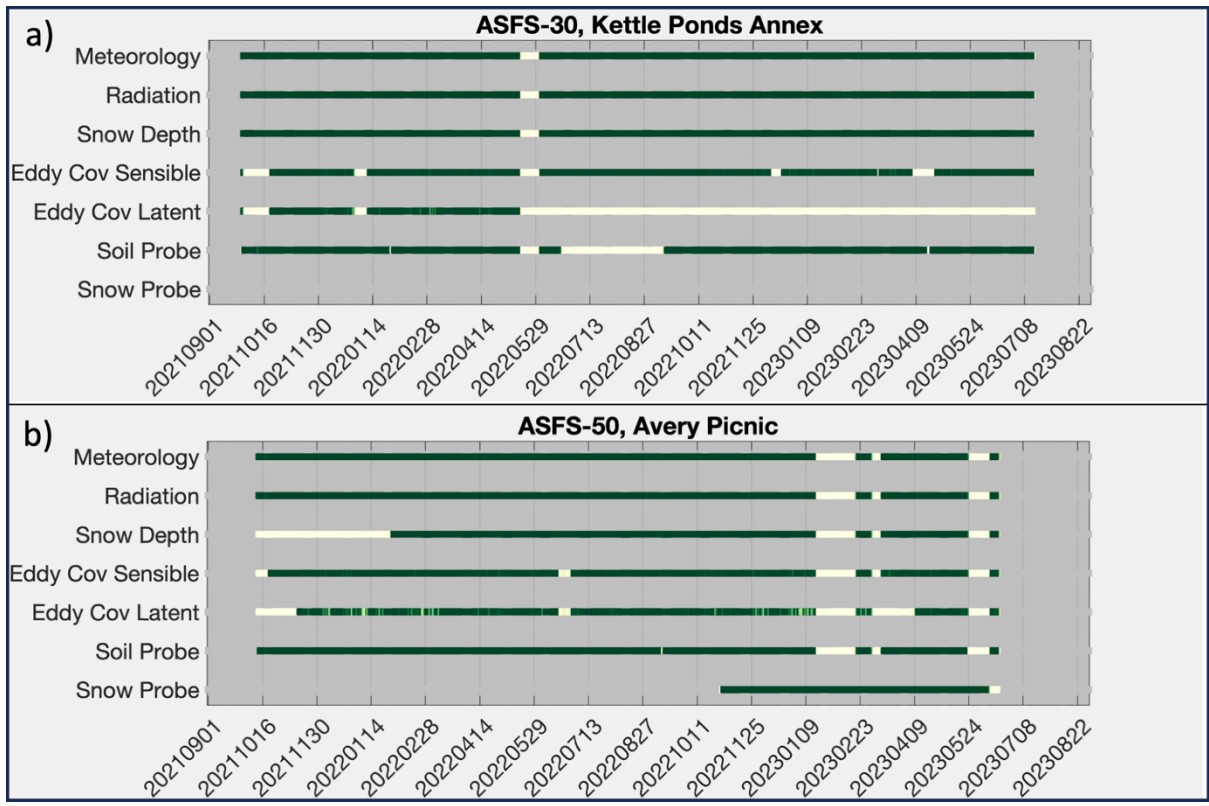

**Figure 3: Time series of daily uptime (% operational) for subsets of the station sensors at (a) Kettle Ponds Annex (KPS-A) and (b) Avery Picnic (AYP). White = 0% uptime and dark green = 100%.**

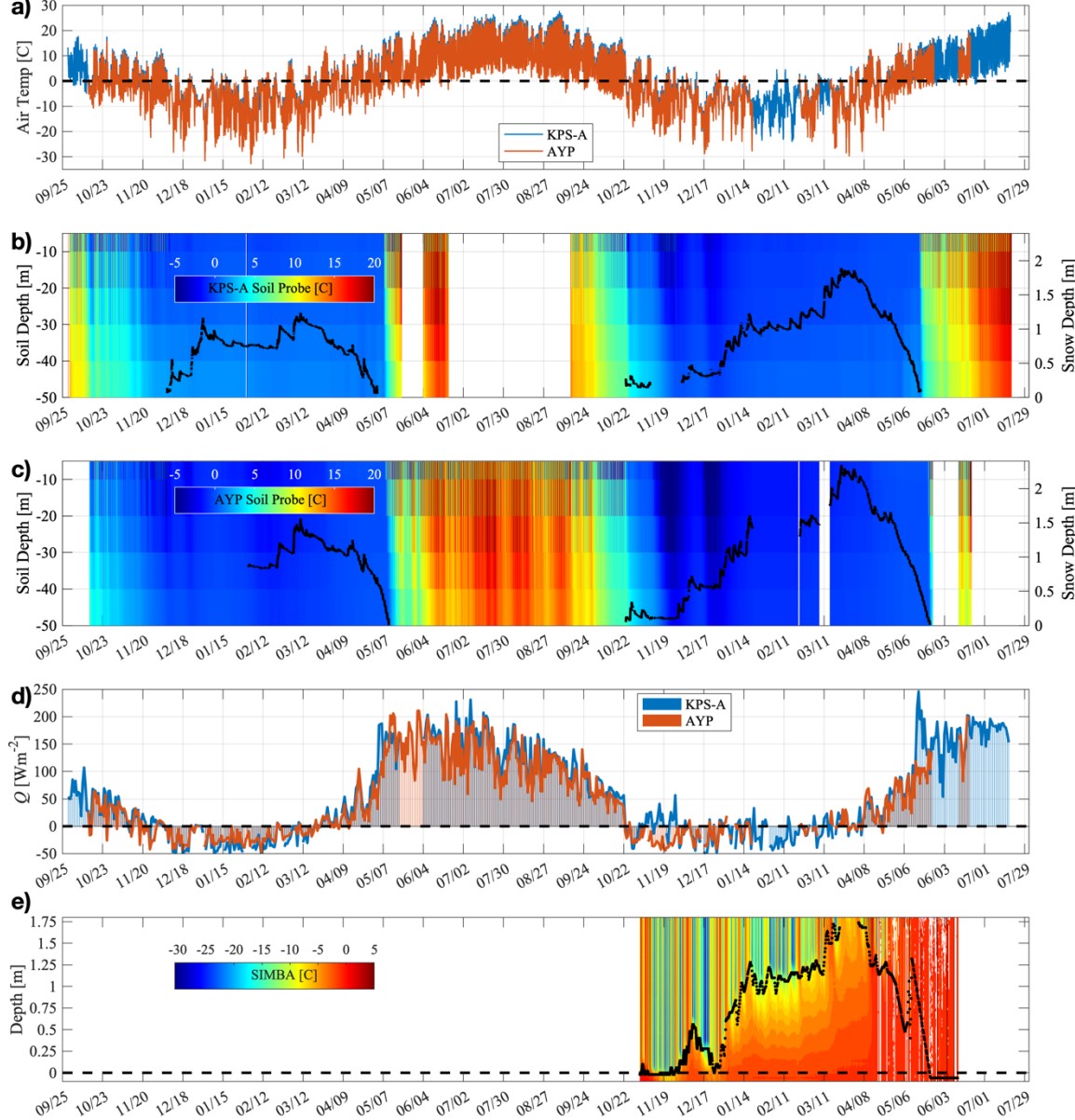


**Figure 4: Sample measurement time series from the ASFS at SPLASH. (a) 10-minute means of air temperature. (b) soil temperature (colours; colormap, left-axis) and snow depth (black; right axis) at Kettle Ponds Annex (KPS-A). (c) as in (b) for Avery Picnic (AYP). (d) net radiation, Q, at both sites. (e) temperatures measures at AYP by the SIMBA thermistor string. Values above the black contour are in air, values below are in snow, values at negative depth (below horizontal dash line) are in soil. The gap in the contour from 22**

**March to 4 April 2023 is the period when the top of the thermistor string was covered by snow. Dates are MM/DD beginning in September 2021 and continuing through July 2023.**




**Figure 5: Comparison of 1-hour mean radiative fluxes between measurements made by ASFS-30 at Kettle Ponds Annex (KPS-A) and those made at Kettle Ponds (KPS) by the NOAA Global Monitoring Laboratory (GML), approximately 400 m upvalley from KPS.**

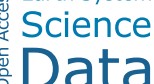

**Figure 6: Planar fits (Wilczak et al, 2001) for Kettle Ponds Annex (KPS-A) (a) and Avery Picnic (AYP) (b). (c) and (d) show respectively for KPS-A and AYP the earth coordinate w' (blue) and the rotated value, W', after planar fit is applied (red). The targets for the distribution are for the mean and 1σ variability of the red distribution to be < the sensor uncertainty of 0.15 m s⁻¹.**




**Figure 7: Comparisons of 1-hour means of $H_s$ in (a), (b), and $H_l$ in (c), (d) between eddy covariance ("ec") and the bulk calculation ("bulk"). Outliers > 4σ were removed from the comparison.**




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
