# Peer review of "Observations of surface energy fluxes and meteorology in the seasonally snow-covered high-elevation East River Watershed during SPLASH, 2021-2023"

_Earth System Science Data, 2024_

## Author Comment (AC1)

We appreciate the constructive and thoughtful feedback provided by the Reviewer, which has helped us to improve the manuscript. Our point-by-point responses are provided below in blue text following each comment, reproduced in black.

**Reviewer 1 Comments (RC1)**

In this manuscript, the authors present and explain the data collected by their Atmospheric Surface Flux Stations (ASFS) during the SPLASH campaign in the East River Watershed in Colorado. They provide thorough background on the surface energy balance which provides sufficient information for understanding the various measurements and data products from ASFS which are described in the remainder of the paper. The authors go into detail on the set up of the sites, the data collection process, and the data processing. Given the information in their manuscript, I was able to easily understand their data sets and believe any scientist could successfully use these data with the guidance of the paper. Additionally, the data sets themselves are high quality, well documented, and sufficiently cleaned/QC'd.

I have a few small recommendations for improvement of the paper:

- In section 5.4, I believe equations 9 and 10 would benefit from a bit more explanation, specifically explaining all parameters in the equations. For example, in equation 9, I was unable to find the meaning of $T_s$; is this the temperature at the surface? Similarly in equation 10, is $z_{soil}$ the depth of soil? This may be common notation, but if the n+1 and n-1 written as superscripts in the numerator of equation 10 are not exponents (i.e. are these time indexes?), it may be beneficial to explicitly note this.

  We agree these equations could be communicated more clearly. Several changes will be made:

    - $T_s$ is the surface temperature that was defined in Eq. (7). We will add a statement beneath Eq. (9) identifying this.
    - In Eq. (10), $z_{soil}$ is the same z as previously stated so we will drop the subscript for consistency.
    - We will make a note in text of the notation for n, as you suggest.
    - We will also add a useful reference to the introduction of these equations, Morris (2018), which reviews this approach alongside direct observations of soil heat flux.

  Morris, S: Variability of ground heat flux at Tiksi station, M.A. Thesis, Department of Geography, University of Colorado, Boulder, Colorado, USA, 82 pp., 2018.

- In Figures 4b and 4c, the units of the Soil Depth are reported in meters with a range of 50 m, whereas in the manuscripts the soil probe is said to be 0.5 m in length. I believe this discrepancy is a simple typo, but if not, some explanation of how these measurements were taken should be added.

  Thank you for catching this. This is indeed a typo in the figure labels and the correct units are of course centimeters. However, to be consistent with the relevant text and the rest of the figure (4e and right-side axes of 4b,c), we will update the figure to keep label as is (in meters) and change the units of the plotted data to meters.

Other minor technical correction: In line 49, SAIL stands for the Surface Atmosphere Integrated *field* Laboratory, please add the world "field".

Thank you for catching this. We will make this change.

---

## Author Comment (AC2)

We appreciate the constructive and thoughtful feedback provided by the Reviewer, which has helped us to improve the manuscript. Our point-by-point responses are provided below in blue text following each comment, reproduced in black.

**Reviewer 2 Comments (RC2)**

This manuscript describes surface flux and meteorology measurement datasets collected during the SPLASH campaign. The measurements were collected at two elevated sites along the East River Watershed valley floor using comprehensive instrument suites. Data files are QC'd and well-documented. The manuscript is well written: the theoretical background in sect. 2 is helpful for context; data processing is described in detail, and the authors properly delineate dataset caveats and workarounds, such as implementing RRTM calculations in resolving flux divergence issues. The discussion about the radiation field and differences between the ASFS and a nearby GML instrument suite is informative and makes general sense (e.g., the effective locality of the ASFS net radiation measurements vs. the GML station). The turbulent fluxes processing is also instructive and makes general sense, as depicted in Fig. 6. That said, I find the Hl approximations (likely the implementation for snow-free conditions) unusable and redundant, overshadowing the rest of this unique dataset. Based on Fig. 7, I wouldn't consider the Hl approximation as "sufficiently well" (l. 417-418). It might be a reasonable approximation for research tasks using 30-day running means or so, in which case the linear fit becomes more relevant, but on a daily or sub-daily scale, considering the absolute values of Hl, this looks inadequate (the range of Hl error is nearly equivalent to its absolute range, suggesting that this approximation is useless in most cases where one wishes to follow the nicely delivered theoretical description in fig. 2). I strongly recommend removing these approximations from the dataset and either revisiting those calculations or simply leaving all the components required for such calculations in the dataset (as I understand is already the case - l. 450-451). I think people could do better science with less good data than abusing bad data. Besides that, the manuscript is well organized and includes easy-to-understand figures, and I only have several minor comments.

Thank you for these comments and feedback on some of the additional work we did to contextualize and verify data, such as the radiative fluxes. Our assessment of the bulk Hl calculation as "sufficiently well" is clearly a subjective statement, which we will remove. The scatter in the comparison between the bulk and eddy covariance shown in Figs 7c,d is large, as you point out. However, for such a comparison, the correlation is high and the mean bias is small, which is why we consider the calculation useful. In a more detailed analysis using a similar bulk algorithm approach over wet and dry soils at a different, more homogeneous mid-latitude site, Grachev et al. (2022 [10.1175/JAMC-D-20-0232.1]) reported similar results to what we find: i.e., that

relatively high correlation and low bias could be achieved, but that the scatter amongst individual samples was large (see e.g., their Fig. 11). As pointed out by Grachev et al., the source of the scatter in the comparison is not necessarily attributable to the bulk approximation, but is significantly influenced by poorly sampled low-frequency contributions to the eddy covariance values that increase the noise of those measurements (e.g., Kessomkiat et al., 2013 [10.1016/j.agrformet.2012.11.019]).

While we did have to make some site-specific adjustments (detailed in the text), the overall approach of making bulk aerodynamic estimates available for community use is consistent with long-standing precedent in our laboratory and collaborations during the period that campaign data sets have been commonly released with DOI (e.g., Cox et al., 2023 [10.1038/s41597-023-02415-5]; Quinn et al., 2021 [10.5194/essd-13-1759-2021]; Bharti et al., 2019 [10.1029/2018JD029761]; Hartten et al., 2018 [10.5194/essd-10-1139-2018]; Miller et al. 2017 [10.5194/tc-11-497-2017]). To confirm, yes the data and methodology necessary to reproduce the bulk values or to create new calculations, can be found in the files. We respect the Reviewer's opinion, but our preference is to retain the bulk calculations in the data set. We have, however, made some changes to the text in response to this comment, replacing

*"...we find that it reproduces eddy covariance measurements of $H_l$ sufficiently well during SPLASH (Figure 7c,d) for inclusion in the data set."*

with

*"The results, shown in Figure 7c,d, exhibit bias and correlative relationships to the eddy covariance measurements comparable to that reported by Grachev et al. (2022) over more homogeneous terrain in the Columbia River basin of Oregon. These statistics include large scatter amongst individual samples, which arises in part from random errors in the eddy covariance measurements, commonly 10-25%, and sometimes larger (Kessomkiat et al., 2013)."*

- Consider adding a table for symbols, abbreviations, and acronyms. The manuscript is full of them, and a table could help readers better orient themselves in the text.

We will include this as Table 1, and it will be called at the end of the Introduction. We have reproduced the new table at the end of this document. The original Tables 1-3 will be renamed 2-4.

- l. 101 - refer directly to relevant panels where the ASFS are shown i.e., paneld d-f in Fig. 1 and d-e in Fig. 2.

Done.

- l. 384 - was --> were

Good catch. Thank you.

- l. 416 - redundant apostrophe.

We aren't sure what punctuation this comment refers to and so we have not made any changes in response.

- l. 434 - remove "To summarize"

Done.

- Fig. 1 caption - Recommend noting explicitly that the ASFS is fully snow-covered in panel d because otherwise it becomes a "Where's Waldo" case...

Instead of referring to panel d in the context of the photo panels e and f, we will change the caption to refer to panel d alongside the map panels a-c, as below:

*Figure 1: The Kettle Ponds Site – Annex (KPS-A). (a) Google Earth$^{TM}$ (Landsat/Copernicus) looking northward (upvalley). (b) and (c) are slope and aspect maps of the KPS-A vicinity (USGS, 2023).* **(d) is a photo of the valley taken facing east from the road depicted in (a). Black dots in (a)-(d) denote the location of ASFS-30. (e)-(f)** *are photos of ASFS-30 in February 2023, June 2023, and September 2021, respectively.*

- Fig. 3 - recommend adding a total uptime percentage for each instrument.

We reported generalized values in the abstract and in Sect. 5.1. It is somewhat subjective to specify a value for what times would be included if the uptime were 100%, which is why we depicted the uptime in graphical form instead. Nevertheless, we have taken your suggestion. We have also adjusted the language in the text and updated the values reported there (when we adjusted the granularity, the rounding came out slightly different).

- Fig. 4—panels b and c—change the left y-axis label units to cm. Panel e—Given that the air temperature is already shown in panel a, I think the air temperature in panel (e) is redundant and somewhat confusing because it is difficult to evaluate the snow temperature with the current depicted color scale. I recommend masking the air temperature and adjusting the color scale only for the snow temperature range.

Thank you for noticing the typo in panels b and c. This was also flagged by Reviewer 1. Instead of changing the y-labels from [m] to [cm], we will change the units of the y-axis from cm to m so as to be more consistent with the right-side axes and the relevant text. We will also make the change you suggest to panel e. The updated figure is reproduced at the end of this document.

Other changes we have made:

- We updated the SOS experiment reference from Gutmann et al. (2023) (a conference presentation) to be Lundquist et al. (2023) (a peer-reviewed article).
- We added the full citation for the Sledd et al. (2024) paper, which was listed as "in revision", but is now published.
- We have reviewed the text for typos and grammar, finding and correcting a few mistakes.
- We have reviewed the formatting of the reference section for consistency with journal requirements, making corrections where necessary, and have checked the DOIs.
- We have updated the acknowledgements, including acknowledging the two anonymous reviewers.

**Table 1. List of acronyms and symbols defined in the text.**

| | |
|---|---|
| **Organizations and Campaigns** | |
| ARM | Atmospheric Radiation Measurement program |
| CIRES | Cooperative Institute for Research in Environmental Sciences |
| DoE | Department of Energy |
| GML | NOAA Global Monitoring Laboratory |
| LBNL | Lawrence Berkeley National Laboratory |
| MOSAiC | Multidisciplinary drifting Observatory for the Study of Arctic Climate |
| NOAA | National Oceanic and Atmospheric Administation |
| PSL | NOAA Physical Sciences Laboratory |
| RMBL | Rocky Mountain Biological Laboratory |
| SAIL | Surface Atmosphere Integrated field Laboratory |
| SOS | Sublimation of Snow experiment |
| SPLASH | Study of Precipitation, the Lower Atmishpere, and Surface for Hydrometeorology |
| **Locations** | |
| AYP | Avery Picnic site |
| GTH | Gothic |
| KPS | Kettle Ponds site |
| KPS-A | Kettle Ponds Annex site |
| UCRB | Upper Colorado River Basin |
| **Models and Sensors** | |
| ASFS-30 | Atmospheric Surface Flux Station station #30 |
| ASFS-50 | Atmospheric Surface Flux Station station #50 |
| HRRR | High-Resolution Rapid Refresh model |
| NWM | National Water Model |
| RRTM | Rapid Radiative Transfer Model |
| SIMBA | Snow Ice Mass Balance Apparatus |
| TDR | Time Domain Reflectometry |
| **Variables and Parameters** | |
| $C$ | Conductive heat flux |
| $H_l$ | Latent heat flux, eddy covariance measurement |
| $H_{lb}$ | Latent heat flux, bulk model calculation |
| $H_s$ | Sensible heat flux, eddy covariance measurement |
| $H_{sb}$ | Sensible heat flux, bulk model calculation |
| $k_{eff}$ | Effective thermal conductivity |
| $LW_D$ | Downwelling longwave radiative flux |
| $LW_U$ | Upwelling longwave radiative flux |
| $Q$ | Net surface radiative flux |
| $q_{sat}$ | saturation specific humidity at the surface |
| $q_{sfc}$ | soil surface specific humidity |
| $R$ | Suface energy balance residual |
| $RH$ | Relative humidity |
| $S$ | Energy storage in the vertical column between the sensor and the surface |
| $S_a$ | Flux divergence in air |
| $S_c$ | Vegetation biomass heat storage |
| $S_g$ | Storage in the subsurface above the measurement of C |
| $S_p$ | Photosynthetic heat storage |
| $S_{snow}$ | Portion of $S_g$ in snow |
| $S_{soil}$ | Portion of $S_g$ in soil |
| $SW_D$ | Downwelling shortwave radiative flux |
| $SW_U$ | Upwelling shortwave radiative flux |
| $S_x$ | Unspecified contributions to S |
| $T$ | Sources of reseidual in the surface energy budget due to heterogeneity |
| $T_s$ | Surface temperature |
| $T_{soil}$ | Soil temperature |
| $VWC$ | Volume Water Content |
| $X$ | Sources of uncertainty contributing to R |
| $\alpha$ | Lee and Pielke (1992) wetness coefficient |
| $\beta$ | Lee and Pielke (1992) wetness coefficient |
| $\varepsilon$ | Surface emissivity |
| $\sigma$ | Stefan-Boltzmann constant (Eq. 7), standard deviation (text) |

[Figure]

Revised Figure 4.